# The impact of different negative training data on regulatory sequence predictions

**Louisa-Marie Krützfeldt**[1,2], **Max Schubach**[1,2], **Martin Kircher**[1,2¤] *

**1** Charité–Universitätsmedizin Berlin, Berlin, Germany, **2** Berlin Institute of Health (BIH), Berlin, Germany

¤ Current address: Charité–Universitätsmedizin Berlin, BIH JRG Computational Genome Biology, Berlin, Germany

\* martin.kircher@bihealth.de

**Data Availability Statement:** All relevant data are within the manuscript and its Supporting Information files.

## Abstract

Regulatory regions, like promoters and enhancers, cover an estimated 5–15% of the human genome. Changes to these sequences are thought to underlie much of human phenotypic variation and a substantial proportion of genetic causes of disease. However, our understanding of their functional encoding in DNA is still very limited. Applying machine or deep learning methods can shed light on this encoding and gapped k-mer support vector machines (gkm-SVMs) or convolutional neural networks (CNNs) are commonly trained on putative regulatory sequences. Here, we investigate the impact of negative sequence selection on model performance. By training gkm-SVM and CNN models on open chromatin data and corresponding negative training dataset, both learners and two approaches for negative training data are compared. Negative sets use either genomic background sequences or sequence shuffles of the positive sequences. Model performance was evaluated on three different tasks: predicting elements active in a cell-type, predicting cell-type specific elements, and predicting elements' relative activity as measured from independent experimental data. Our results indicate strong effects of the negative training data, with genomic backgrounds showing overall best results. Specifically, models trained on highly shuffled sequences perform worse on the complex tasks of tissue-specific activity and quantitative activity prediction, and seem to learn features of artificial sequences rather than regulatory activity. Further, we observe that insufficient matching of genomic background sequences results in model biases. While CNNs achieved and exceeded the performance of gkm-SVMs for larger training datasets, gkm-SVMs gave robust and best results for typical training dataset sizes without the need of hyperparameter optimization.

## Introduction

Regulatory sequences play an important role in the control of transcription initiation. Variants in regulatory elements can lead to changes in gene expression patterns and are associated with various diseases [1–3]. Deciphering the encryption of regulatory activity in genomic sequences is an important goal and an improved understanding will inevitably contribute to a better interpretation of personal genomes and phenotypes. While available approaches for measuring

**Funding:** The author(s) received no specific funding for this work.

**Competing interests:** The authors have declared that no competing interests exist.

changes in regulatory sequences activity in a native genomic context are still very limited in their throughput [4], machine learning methods can be applied for regulatory activity prediction directly from DNA sequence and reveal enriched sequences patterns and arrangements [5].

With some notable exceptions [6,7], there is a strong link between transcription factors (TFs) binding to regulatory elements and general DNA accessibility, i.e. open chromatin. While the screening of individual TFs is tedious and restricted by the availability of appropriate antibodies, chromatin accessibility can be measured genome-wide and in multiple assays (e.g. DNase-seq, ATAC-seq or NOMe-seq). DNase I hypersensitive site sequencing (DNase-seq) provides a gold-standard for the detection of chromatin accessibility [8] and is widely used by the ENCODE Consortium as a sensitive and precise reference measure for mapping regulatory elements [9,10]. It allows the detection of active regulatory elements, marked by DNase I hypersensitive sites (DHS), across the whole genome [11,12].

Machine learning approaches identify regulatory elements among other coding or non-coding DNA sequences based on structured patterns of their DNA sequences. Many of these patterns can be matched to known transcription factor binding sites (TFBSs) [13,14] and their relative orientation and positioning. TFs are known to have different binding affinities to DNA sequences, to bind preferentially to a specific set of short nucleotide sequences named binding motifs [13] and to vary in their cell-type expression [9]. Further, TFs can have preferences for a three-dimensional structure of the DNA [14]. While DNA structure can be predicted from the local sequence context, the same DNA shape can be encoded by different nucleotide sequences. There are probably additional patterns, but GC-related sequence features are commonly identified as predictors of regulatory activity and can affect nucleosome occupancy due to differential DNA binding affinity of histone molecules [15].

Gapped k-mer support vector machines (gkm-SVMs) [16–18] and convolutional neural networks (CNNs) [19–21] have been recently applied in multiple studies to either predict regulatory activity/function or to identify key elements of the activity-to-sequence encoding. Support vector machines are a class of machine learning algorithms mainly used for classification problems, where they find a projection in a high-dimensional space with an optimal decision boundary (hyperplane). In the case of gkm-SVMs, SVM classifiers are applied to strings representing gapped k-mers, i.e. oligonucleotides of fixed size $k$ from a library of DNA sequences that includes non-informative positions named gaps. A convolutional neural network is a class of deep neural networks that takes advantage of hierarchical patterns in data and assembles complex patterns (using multiple convolution and pooling layers) from simpler, smaller patterns, so-called convolutional kernels. While DHS datasets serve as positive training data for these machine learning algorithms, the ideal composition of the negative training dataset is still an unsolved question. There are two commonly used approaches for the generation of negative training data, the selection of sequences from genomic background [18] and k-mer shuffling of the positive sequences [22–24].

In case of genomic background sequences, the negative training dataset is composed of sequences from the genome that are not overlapping DHS regions. However, using non-DHS regions does not guarantee selecting only inactive sequences, due to incomplete sampling of the cell-type under consideration or activity in other cell types. Typically, when selecting background sequences certain properties of the positive training set, e.g. sequence length and repeat fraction, are preserved. Due to this matching of sequence features, this method can be computationally expensive. An alternative approach, k-mer shuffling, is computationally efficient and generates synthetic DNA sequences. A collection of negative sequences according to this approach is composed of the shuffled DHS sequences while preserving each original sequence' k-mer counts.

Our work investigates the choice of the negative training dataset and its impact on model performance for predicting regulatory activity from DNA sequences. By applying gkm-SVM and CNN models, both machine learning methods and approaches for negative training data generation are compared. Models are trained on DHS regions from experiments in five different cell lines and various matching negative sets. Performance of the resulting models is evaluated on three different tasks. The first task is the binary classification of DNA sequences into active and inactive for the specific cell line, i.e. classical hold-out performance for individual DHS datasets. The second task tests the ability to learn tissue-specificity and evaluates performance in identifying cell-type specific DHS sequences. In the third task, models are applied to the prediction of enhancer activity and evaluated on an experimental dataset of activity read-outs from a reporter assay [25].

We show a large impact of the negative training dataset on model performance. Models trained on highly shuffled sequences perform worse except for hold-out performance, while models trained on genomic sequences excel on the more complex tasks of tissue-specific activity prediction and quantitative activity prediction. We speculate that models trained on sequence shuffles learn features of artificial sequence rather than regulatory activity. We also note that insufficient matching of selected genomic background sequences may result in model biases. While CNN performance was improved and exceeded gkm-SVMs for larger training datasets, gkm-SVMs gave better results for small training dataset sizes.

## Materials and methods

### Training, validation and test data

In general, positive and negative sequences (except for the independent liver enhancer dataset, see below) were split into three datasets for training, validation, and testing. The validation (hyperparameter optimization) and test sets (performance evaluation) were chromosome hold-out sets of chromosomes 21 and 8, respectively. Training was performed on sequences located on the remaining autosomes and gonosomes.

### Positive training data: DNase I hypersensitive (DHS) data

DNase-seq datasets were used as positive datasets for regulatory sequence prediction. Seven DNase-seq datasets (narrow peak calls) from experiments in five different cell lines (A549, HeLa-S3, HepG2, K562, MCF-7) were downloaded from ENCODE. Multiple technical replicates were merged into one file per experiment, combining overlapping (minimum of 1 bp) or adjacent sequences into a single spanning sequence. For cell lines A549 and MCF-7 two pooled DHS datasets exist (S1 Table), we refer to those as experiments A and B. We determined the overlap of merged peak sets across experiments in the same cell-type and across cell-types. For peaks to be considered overlapping between datasets, we required a 70% overlap in their coordinate ranges. We calculated pairwise overlap as number of overlapping peaks divided by the number of peaks in the union of both data sets. DHS regions were defined 300 bp around the center of the narrow peaks/merged segments and reference genome sequences used (GRCh38 patch release 7, GRCh38.p7). Sequences located on alternative haplotypes, on unlocalized genomic contigs, or containing non-ATCG bases were excluded. An overview of the used DNase-seq datasets is presented in S1 Table.

### Negative training data: Genomic background data and k-mer shuffling

To obtain genomic background sequences as negative training datasets, DNA sequences with matching repeat and GC content (as in the DHS set) were randomly selected from the genome.

While matching repeat content is supposed to correct for potential alignment biases, GC matching is performed to compensate for potential biases caused by better experimental recovery of high GC sequences in DNA handling. Datasets were generated using the genNullSeqs function of the R package gkmSVM [17]. For this purpose, genome sequences (GRCh38.p7) were obtained from UCSC and stored in Biostrings [26] BSgenome.Hsapiens.UCSC.hg38.masked [27]. To make sure that matching sequences were found for at least 80% of the samples in each dataset, the batch size and maximum number of trials were increased (batch-size = 10000, nMaxTrials = 100). The tolerance for differences in repeat ratio and relative sequence length were set to 0, but the tolerance for differences in GC content was varied for different training datasets ($t_{GC}$ = {0.02, 0.05, 0.1}). We explored a range of GC tolerances as GC content has been associated with gene expression regulation and the original study [17] did not explore this parameter.

To generate neutral DNA sequence for the negative training dataset, positive sequences were shuffled while preserving the k-mer counts. Here, k-mer shuffling datasets were generated using fasta_ushuffle (https://github.com/agordon/fasta_ushuffle, accessed 02/26/2020), a wrapper for the fasta file format to uShuffle [28]. The parameter k which indicates the size of the preserved k-mers was varied for different datasets ($k$ = [1,7]). Here, we explore a range of k as we were unable to identify a consensus value from prior literature [22,29–32]. For each positive sequence, 200 shuffled sequences were generated and the sequence with minimal 8-mer overlap to the respective positive sequence chosen.

### Tissue-specific test data

Assessing the capability of models to predict tissue-specific regulatory activity, datasets with tissue-specific DHS regions were used for testing. For each of the five cell lines, one positive and one negative dataset was generated. For A549 and MCF-7, experiments B were chosen based on best hold-out performance of the gkm-SVM model (shuffled, $k$ = 2). Positive datasets contain non-overlapping DHS regions to the other four cell lines. The corresponding negative datasets contain DHS regions of the other four cell lines not overlapping with DHS regions of the cell line under consideration. A maximum 30% overlap of regions was tolerated. Tissue-specific datasets were not used for training, but split up in validation and test (i.e. chromosome hold-out sets of chromosomes 21 and 8, respectively; S2 Table) to exclude overlaps with model training.

### Liver enhancer activity data

Models were tested on an independent dataset of experimental activity readouts [25] to evaluate the models' ability to quantitatively predict enhancer activity. The underlying Massively Parallel Reporter Assay experiments were performed in HepG2 cells infected with lentiviral reporter constructs bearing candidate enhancer sequences chosen on the basis of ENCODE HepG2 chromatin immunoprecipitation sequencing (ChIP-seq) peaks for *EP300* and H3K27ac marks. We used $\log_2$ RNA/DNA ratios reported for the wild-type integrase experiments and excluded control/synthetic sequences. GRCh37 sequence coordinates were converted to GRCh38.p7 and regulatory sequences where coordinate liftover changed the fragment length were excluded (1 out of 2236). The original fragment size of 171 bp was extended on both ends to a total of 300 bp.

### Merged datasets of different sizes

A total of six DHS datasets of different sizes from a mixture of the five cell lines were created. 100k or 120k DHS regions from each cell line were randomly chosen and resulted in datasets

of 500k or 600k DHS regions, respectively. Derived from the 500k dataset, smaller datasets (50k, 100k, 200k and 350k) were randomly sampled.

## Gapped k-mer support vector machine (gkm-SVM)

Gkm-SVM models were trained with default parameters (word length $l$ = 10, informative columns $k$ = 6) and a weighted gkm kernel, as these parameters were previously used for regulatory sequence prediction [18]. To handle big training datasets, the R package LS-GKM [17,33] was used.

## Convolutional neural network (CNN)

Two different CNN architectures were used. The first architecture, named 4conv2pool4norm (according to 4 convolutional layers, 2 max-pooling layers and 4 normalization layers), was previously presented as DeepEnhancer for accurate prediction of enhancers based on DNA sequence [34]. A smaller network named 2conv2norm (according to 2 convolutional layers and 2 normalization layers), was derived from the 4conv2pool4norm network. Architecture and layer properties of networks are described in S3 and S4 Tables.

Models were trained in the Python deep learning library Keras based on the tensorflow interface [35]. The Adam optimizer [36] was used with default parameters as previously suggested [37]. In addition to the default parameters for batch size (200) and learning rate (0.001), a different parameter set was examined (batch size = 2000, learning rate = 0.0002). For both architectures, the higher batch size and lower learning rate were chosen based on accuracy and standard deviation on the validation set (chromosome 21 hold-out, regulatory activity task). Models were trained over 20 epochs showing a convergence of the estimated loss on the validation sets and no signs of overfitting (see S1 and S2 Figs). Network training was repeated 10 times using different seeds. For regulatory activity and tissue-specific activity prediction, one out of the 10 models was chosen for further analysis based on median model performance (chromosome 21 hold-out).

## Evaluation tasks and model evaluation

Each model was evaluated on three tasks and different performance measures were chosen depending on the task. Receiver Operating Characteristic (ROC) curve and area under ROC curve (AUROC) values are commonly used and a good measure if test datasets are balanced between classes [38] and if the confidence in class labels is similar. An alternative method for imbalanced datasets are Precision-Recall (PR) curves. In contrast to AUROC, area under PR curve (AUPRC) depends on the imbalance of the dataset [39]. A perfect model has an AUPRC value of 1, a random model an AUPRC value equal to the proportion of positive samples in the test set. The R packages PRROC [40,41] and pROC [42] were used to calculate the respective values.

For task one (regulatory sequence prediction), AUROC, AUPRC and recall values were used for model evaluation. First models were tested on validation sets to identify best parameters for generating the negative training set based only on recall measures. Based on the test sets, performance of models trained on genomic background or shuffled sequences were compared for each classifier. We evaluated models on their respective hold-out sets. Additionally, models trained on shuffled data were evaluated on hold-outs using genomic background sequences as negative sets. Pairwise comparisons of model performance were realized by Wilcoxon signed-rank tests.

The second task considered the models' tissue-specificity. Again, negative training dataset parameters were chosen according to validation dataset performance. Classifiers and types of

negative training sets were then compared based on the test datasets. To assess the model performance on task 2 (tissue-specific prediction), PR and ROC curves and corresponding AUPRC and AUROC values were used.

For the third task, models were tested on a regression problem and used to predict activity of liver enhancer sequences for which experimental readouts were previously published [25]. Here, Spearman rank correlations were calculated between prediction scores and available $\log_2$ activity ratios.

## Transcription factor (TF) binding motif analysis

Training dataset sequences were searched for known TF binding profiles and for each dataset the number of matched motifs per 300 bp calculated. A set of 460 non-redundant profiles derived from human TFBSs was exported from the JASPAR CORE database [43]. Profile matches were identified using FIMO [44] with default parameters and a maximum number of motif occurrences retained in memory of 500,000.

## Frequency distribution of 8-mers

All potential 8-mers consisting only of nucleotides A, C, G and T were extracted from all 24 major chromosomes of the human reference genome sequence (GRCh38) with their absolute count. Non-zero counts were obtained for all 65,536 possible 8-mers. Counts were Z-score transformed, i.e. mean-centered and the standard deviation normalized to 1. Potential 8-mers were further extracted from test sequences and the Z-score of their genomic frequency looked up. We also looked up Z-scores for the top 100 scoring 8-mer sequences for each of 128 kernels in the first convolutional layer of the CNN models.

## GC content distribution

The GC content distribution was calculated for active DHS regions in HepG2, three corresponding genomic background datasets with varied GC content tolerance and random genomic sequences. One million random sequences of length 300 bp were selected from GRCh38. p7 (excluding alternative haplotypes and unlocalized contigs) as a reference for the composition of the human genome. For each sequence, GC content was calculated using the R package 'seqinr' [45].

# Results

## Training models for regulatory activity prediction

To investigate the performance of machine learning methods for regulatory activity prediction from DNA sequence and the impact of negative data set composition, multiple models were compared. Two machine learning approaches, gkm-SVMs and CNNs with two different architectures, were used. The CNN architectures were derived from DeepEnhancer [34] and are referred to as 2conv2norm and 4conv2pool4norm (see Methods). Each model was trained on a positive dataset of DHS regions in a specific cell line (active regulatory sequences) and a corresponding set of negative sequences. Negative training datasets were generated using two different approaches (genomic background, k-mer shuffles) and variation of parameters led to ten different negative training sets per positive dataset. In the genomic background approach three different GC content tolerances ($t_{GC}$ = {0.02, 0.05, 0.1}) were tested. In the k-mer shuffling approach, the size of the preserved k-mers varied from 1 to 7. The influence of the negative training dataset on model performance was evaluated on chromosome hold-out validation and test sets. First, model hyperparameters were selected on the validation sets, then

the models' capability to predict (tissue-specific) regulatory activity was assessed on the test sets, as well as from a quantitative prediction of enhancer activity on an independent experimental dataset.

## Model performance on chromosome hold-out sets

To measure model performance, we calculated ratios of correctly predicted positive samples, i.e. recall and the area under precision recall curve (AUPRC). For each classifier, we chose one model trained on genomic background and one model using k-mer shuffles for further experiments. To select these models, we compared their performance on a hold-out set of active DHS regions on chromosome 21 (validation set). Since we did not observe relevant effects for parameters of the genomic background set (S3 and S4 Figs), we chose the most stringent parameter ($t_{GC} = 0.02$). In contrast, when comparing models trained on shuffled sequences, model performance depended on the size of preserved k-mer $k$ (S5 and S6 Figs), with small $k$ resulting in better performance and high $k$ falling behind the genomic background sets. We note that the value of $k$ is anticorrelated to the number of known transcription factor binding site (TFBS) motifs remaining in the negative training sequences (S7 Fig) and suggests that models may identify positive samples based on TFBS frequency. While models with $k = 1$ show the best results, we chose $k = 2$ as shuffled sequences preserving dinucleotide composition are widely used [22].

Selected models were then compared across classifiers on a second chromosome hold-out dataset (chromosome 8, test set). In accordance with previous studies, CNNs and gkm-SVM classifiers are both able to predict active DHS regions from the hold-out sets with high recall and AUPRC values (S8 Fig). We do not see a clear difference between the two CNNs tested. However, models trained on highly shuffled data perform significantly better than models trained on genomic background data.

Fig 1 represents AUROC values for all selected models tested on hold-out sets including genomic background sequences (top panels) or shuffled sequences as negative test sets (bottom panels). Differences between CNN and gkm-SVM classifiers are marginal in this comparison and models perform best on the composition that they were trained on. This is in line with models relying on features from both negative and positive sequences. However, models trained on shuffled sequences show a larger drop when tested on a test set using natural sequences as negative class. For example, gkm-SVM models trained on shuffled sequences drop from a mean AUROC of 0.96 to 0.64, while models trained on natural sequences drop from a mean AUROC of 0.90 to 0.83. This suggests that model training may focus more on the shuffled sequences in this case.

To explore further, how models were influenced by the negative sets, we analyzed 8-mer frequency in the different test data set classes (i.e. DHS sites, genomic background, and shuffled sequences) as well as 8-mers prioritized in the first convolutional layer of our CNN models. We compared these 8-mers based on their genomic frequency across the human genome. We observe that 8-mers in the genomic background negative sets are on average more frequent than 8-mers from DHS sites (positive sets) and those are more frequent than 8-mers from shuffled negative sequences (S10A Fig). While effects are more subtle, similar effects propagate into 8-mers identified in the first convolutional layers (S10B and S10C Fig), with models trained on genomic background sequences learning to identify more common 8-mers (Wilcoxon rank tests, $p < 2.2e^{-16}$). Consequently, rare motifs in shuffled negative sequences are learned by these models and may negatively impact model performance.

For A549 and MCF-7 cell lines with two available DHS sets from ENCODE, two separate models were trained and their performance on the test sets compared among all cell

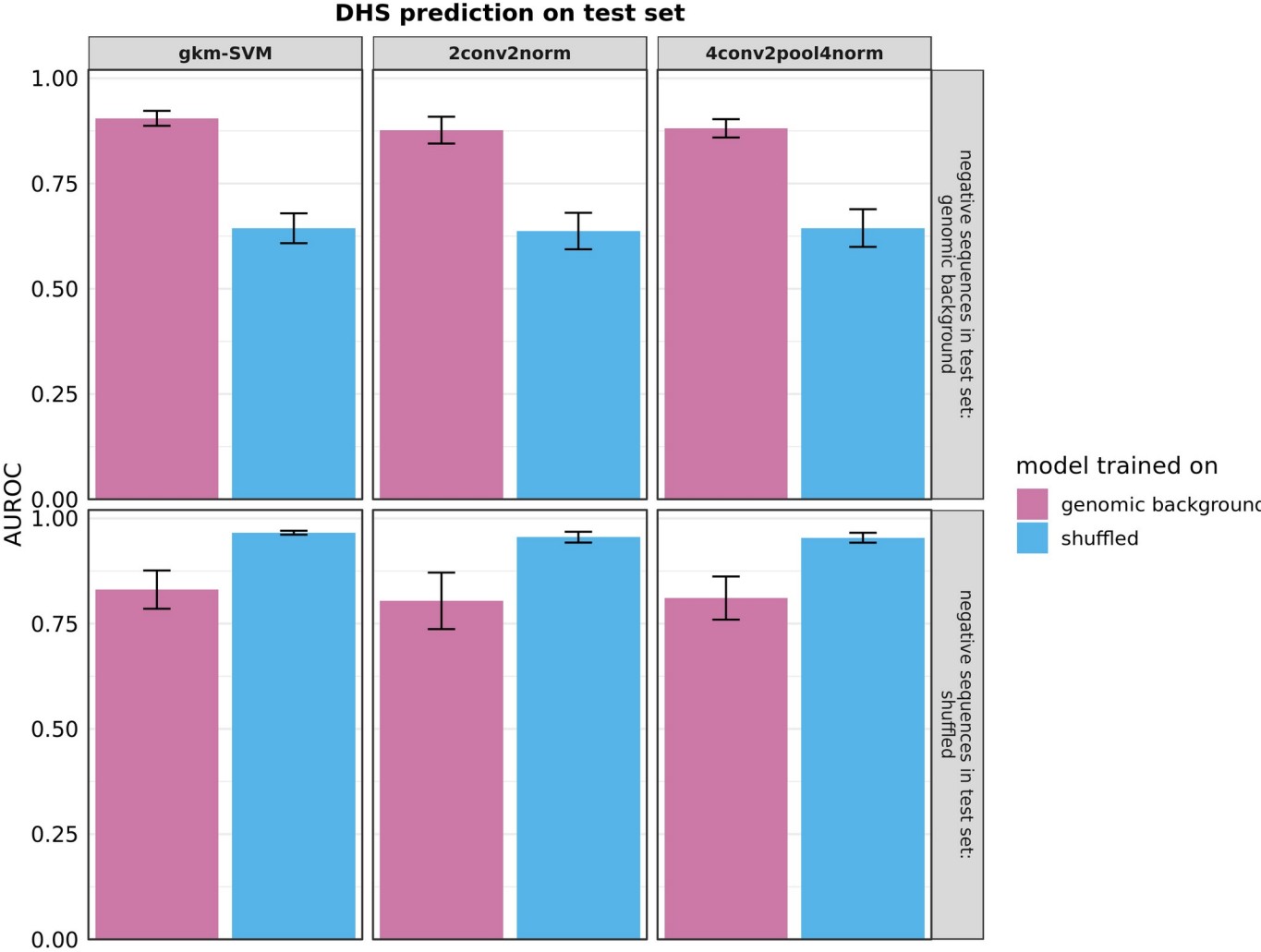

**Fig 1. AUROC values for regulatory sequence prediction.** Models were trained on sequences of DHS regions (positive) with corresponding sets of negative sequences. For each classifier two different negative training sets are compared; sequences were either chosen from genomic background ($t_{GC} = 0.02$) or generated by shuffling positive sequences and preserving k-mer counts (k = 2). Models were tested on a chromosome 8 hold-out test set. The top panels show the results for testing on hold-out sets using genomic background sequences as negative sets, the bottom panels show the results for testing on hold-out sets using shuffled sequences as negative sets. AUROC values were calculated to compare model performance. Seven models were trained on data derived for specific cell lines, bars represent the mean and error bars the standard deviations across models.

lines (Table 1). We see that performance generalizes well across diverse cell lines (e.g. breast, cervix, lung, liver cancer and leukemia), suggesting that organismal rather than tissue-specific active regulatory regions are predicted. As an example, Table 1 shows recall values for the gkm-SVM models trained on shuffled sequences (k = 2) ranging from 0.79 to 0.88 for other cell types. This highlights the ability of models to generalize in identifying potential open chromatin regions, despite DHS peaks in training, validation and test showing very little overlap across cell-types (pairwise overlap of ≤ 0.34) or even across experiments of the same cell-type (pairwise overlap of up to 0.53, S11 Fig). As to be expected, models trained on A549 training sets perform best on A549 test sets (recall of 0.86 and 0.88, respectively) and MCF-7 models perform best on MCF-7 datasets (recall of 0.90 and 0.91, respectively).

**Table 1. Recall of test set regulatory sequence prediction for different cell lines.** Gkm-SVM models were trained on DHS datasets (positive) and corresponding negative sets of k-mer shuffled sequences (k = 2, k = 7) or genomic background sequences ($t_{GC}$ = 0.02) for A549 or MCF-7 cells; cell lines with two training datasets (A/B) each. Model performance was evaluated based on recall for hold-out sets (chromosome 8). There are seven different hold-out sets derived from different cell lines and we assess model generalization across cell-types. Best performance is observed for models trained on highly shuffled sequences (k = 2), model performance is reduced when trained on genomic background, while the performance of models trained on lightly shuffled sequences (k = 7) is considerably worse. Datasets are named according to S1 Table. Results for the CNN models (2conv2norm and 4conv2pool4norm) are available in S5 and S6 Tables, respectively.

| | | Model | | | | | | | | | | | |
| --- | --- | --- | --- | --- | --- | --- | --- | --- | --- | --- | --- | --- | --- |
| | | Shuffled (k = 2) | | | | Shuffled (k = 7) | | | | Genomic background ($t_{GC}$ = 0.02) | | | |
| | | A549 (A) | A549 (B) | MCF-7 (A) | MCF-7 (B) | A549 (A) | A549 (B) | MCF-7 (A) | MCF-7 (B) | A549 (A) | A549 (B) | MCF-7 (A) | MCF-7 (B) |
| Recall (test set) | A549 (A) | 0.896 | 0.863 | 0.873 | 0.859 | 0.599 | 0.495 | 0.546 | 0.497 | 0.838 | 0.764 | 0.778 | 0.746 |
| | A549 (A) | 0.882 | 0.880 | 0.855 | 0.846 | 0.547 | 0.520 | 0.514 | 0.484 | 0.820 | 0.817 | 0.769 | 0.755 |
| | HeLa-S3 | 0.877 | 0.852 | 0.863 | 0.848 | 0.502 | 0.423 | 0.486 | 0.433 | 0.779 | 0.725 | 0.752 | 0.719 |
| | HepG2 | 0.838 | 0.822 | 0.813 | 0.799 | 0.436 | 0.382 | 0.410 | 0.377 | 0.703 | 0.666 | 0.653 | 0.627 |
| | K562 | 0.834 | 0.802 | 0.809 | 0.793 | 0.491 | 0.391 | 0.433 | 0.399 | 0.688 | 0.611 | 0.627 | 0.607 |
| | MCF-7 (A) | 0.872 | 0.844 | 0.905 | 0.893 | 0.526 | 0.438 | 0.618 | 0.562 | 0.794 | 0.737 | 0.868 | 0.842 |
| | MCF-7 (B) | 0.870 | 0.853 | 0.906 | 0.900 | 0.556 | 0.473 | 0.636 | 0.604 | 0.811 | 0.762 | 0.882 | 0.872 |

### Prediction of tissue-specific regulatory sequences

As seen in the previous experiments, models trained on data derived from one cell line may generalize in predicting active DHS regions in other cell lines. While some regulatory sequences are active in multiple cell types, others are specifically active in only one cell type. To further assess the models' capability to predict tissue-specific regulatory activity, we used datasets containing tissue-specific DHS sequences for further testing. We selected DHS sequences only active in the training cell line (positive samples) and DHS regions not active in this cell line but active in at least one of the other cell lines (negative samples).

Again, we first tested parameter choice on a validation set (chromosome 21 hold-out). Since HeLa-S3 models performed best, we focus the presentation of results on this cell line For the genomic background set, we chose again the most stringent parameter ($t_{GC}$ = 0.02) as models trained using genomic background showed similar performance independent of the GC content tolerance (S12 Fig). For shuffled sequences, we picked k = 7 based on precision recall (S13 Fig). Model performance tends to increase with higher size of preserved k-mers in shuffled sequences. The high value of k preserves a number of TFBS motifs (46±2 motifs per 300 bp) similar to the positive set (47±2 motifs per 300 bp, S7 Fig), suggesting that presence of tissue-specific factors as well as relative positioning may be most critical for model performance. We notice that performance is considerably reduced compared to the first task and see big differences regarding model performance across different training cell lines (S7 and S8 Tables).

We present HeLa-S3 models for the final evaluation on the hold-out test set (chromosome 8). Fig 2 shows ROC and PR curves for 2conv2norm (Fig 2A and 2B), 4conv2pool4norm (Fig 2C and 2D) and gkm-SVM (Fig 2E and 2F) models. Predicting tissue-specific regulatory activity, the performance of models is low, but models trained on genomic background data generally perform better than models trained on shuffled sequences (e.g. AUROC differences of 6.7/6.8% for the two different CNN architectures). We do not measure a clear performance difference between the two different CNN architectures, but observe that the gkm-SVM model performed a bit better (AUROC +2%) on this task.

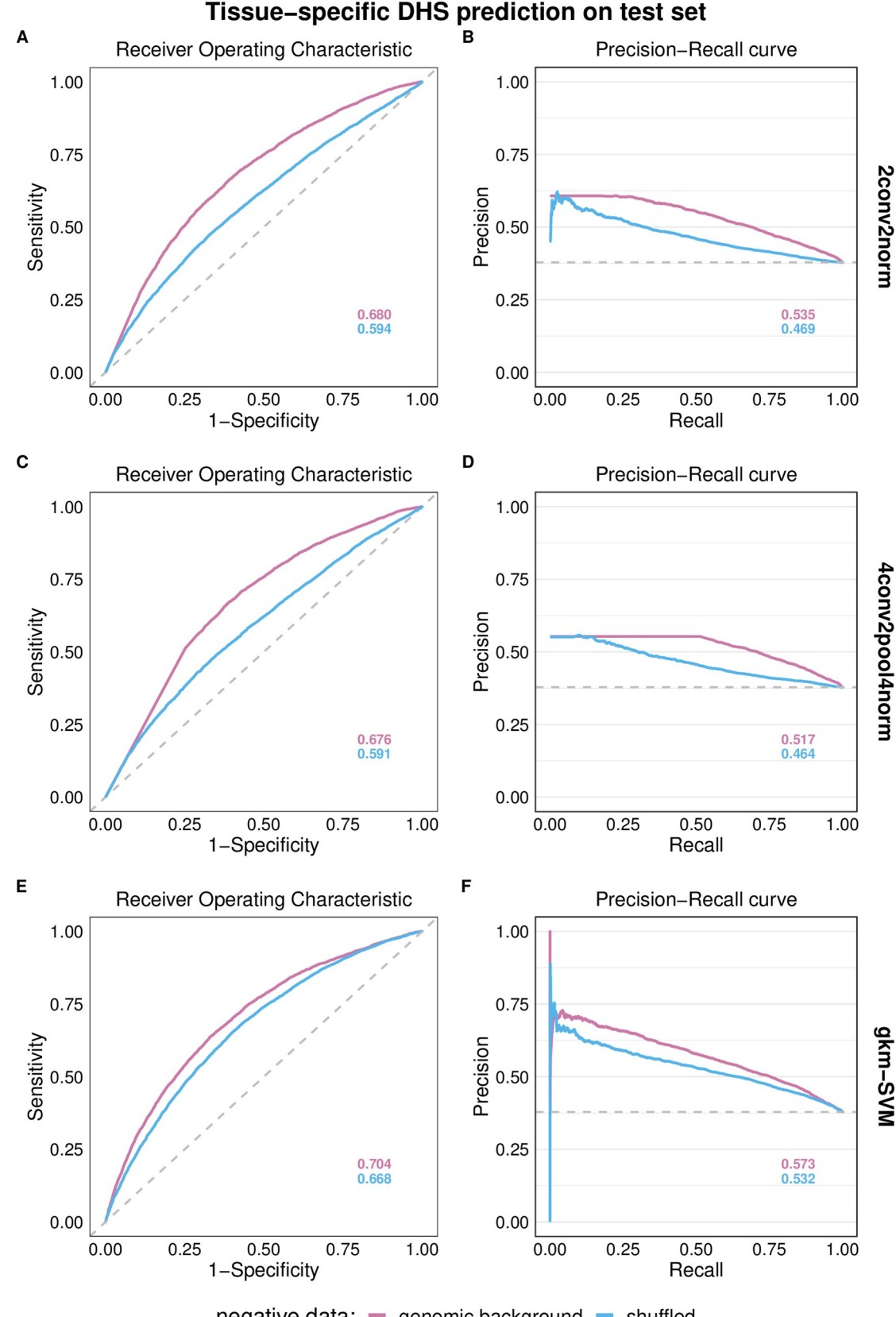

**Fig 2. HeLa-S3 model performance for tissue-specific regulatory sequence prediction.** Models were trained on sequences of DHS regions active in HeLa-S3 cells (positive) and negative sequence sets of either matched genomic background sequences ($t_{GC}$ = 0.1) or k-mer shuffled *(k = 7)* sequences. Models were tested on DHS sequences only active in HeLa-S3 (positive) and DHS sequences active only in one or multiple other cell lines (A549, HepG2, K562, MCF-7) (negative). Dashed lines represent random model performance. Panels (A) and (B) show ROC and PR curves for 2conv2norm models, (C) and (D) show ROC and PR curves for 4conv2pool4norm models, (E) and (F) show ROC and PR curves for gkm-SVM models. Corresponding AUROC and AUPRC values are provided.

## Quantitative enhancer activity prediction

Lastly, we evaluated the models capability of predicting quantitative enhancer activity for an independent experimental dataset. For this purpose, we used enhancer activity readouts from published data [23] and calculated Spearman correlation of predicted scores with known activity readouts.

Since enhancer activity was measured in HepG2 cells, we first applied our models trained on HepG2 DHS data. In contrast to earlier results, model performance differs across models trained using different GC content matching of the genomic background datasets. Models trained on sequences that varied most from positive sequences regarding their GC content, performed best (S14 Fig). Therefore, this less stringent matching parameter was considered here. Next, the shuffling parameter *k* was evaluated on enhancer activity prediction for HepG2 models. Here, the extremes, i.e. models trained on highly shuffled sequences (*k* = 1) or models with low shuffling (*k* = 7) performed worse for the different model types (S15 Fig). Best performance is achieved for *k* = {3,4} for gkm-SVM, while for the CNN architectures *k* = {5,3} perform best. Based on these results, the parameter *k* = 3 was chosen. Inoue et al. [25] analyzed how their enhancer activity readouts correlated with a gkm-SVM model trained from more than 200,000 ENCODE ChIP-seq peaks observed in HepG2 [17]. Our HepG2 models based on DHS sites did not achieve the performance of a Spearman's ρ of 0.28 reported before [25] (see Fig 3). Therefore, other cell-type models were also tested and A549, HeLa-S3 and K562 models achieved or exceeded the reference performance (Fig 3 incl. HepG2 and K562, further cell-types see S16 Fig). Compared to others, the HepG2 training set is smaller (123k compared to 281k HeLa-S3, 222k K562 and 192k A549, S1 Table). To investigate whether the size of the training dataset influences model performance, new models were trained on datasets of varying size (50k to 600k), by sampling sequences from all cell lines (see Methods). We note that sampling across cell lines dilutes a tissue-specific signal and we expect that correlation with experimental readouts might be reduced.

Again, we evaluated the correlation of prediction scores and activity readouts. Results are presented in Fig 4. Model performance of gkm-SVM classifier seems very stable across training set sizes and repeated training runs, but due to runtime (more than 3 weeks) we did not test more than 350,000 positive training examples. Using genomic background sequences clearly outperformed shuffled sequences. For CNNs, the more complex architecture (4conv2pool4-norm) outperformed 2conv2norm on both negative sets. To achieve or exceed the gkm-SVM performance, 4conv2pool4norm required larger training datasets (6-7x more data). Looking across 10 trained CNN models per data set, we see considerable variance in model performance, suggesting high stochasticity in training, likely originating from non-optimal parameters (e.g. batch size, learning rate, convergence). Gkm-SVM (0.29) and 4conv2pool4norm models (0.30) both exceeded the reference Spearman's ρ value (0.28, Fig 4), despite effects of pooling training datasets across cell lines.

## Discussion

We found that CNN models and gkm-SVM models are equally suited for active DHS prediction. While similar in performance, CNN models showed larger variance across training runs

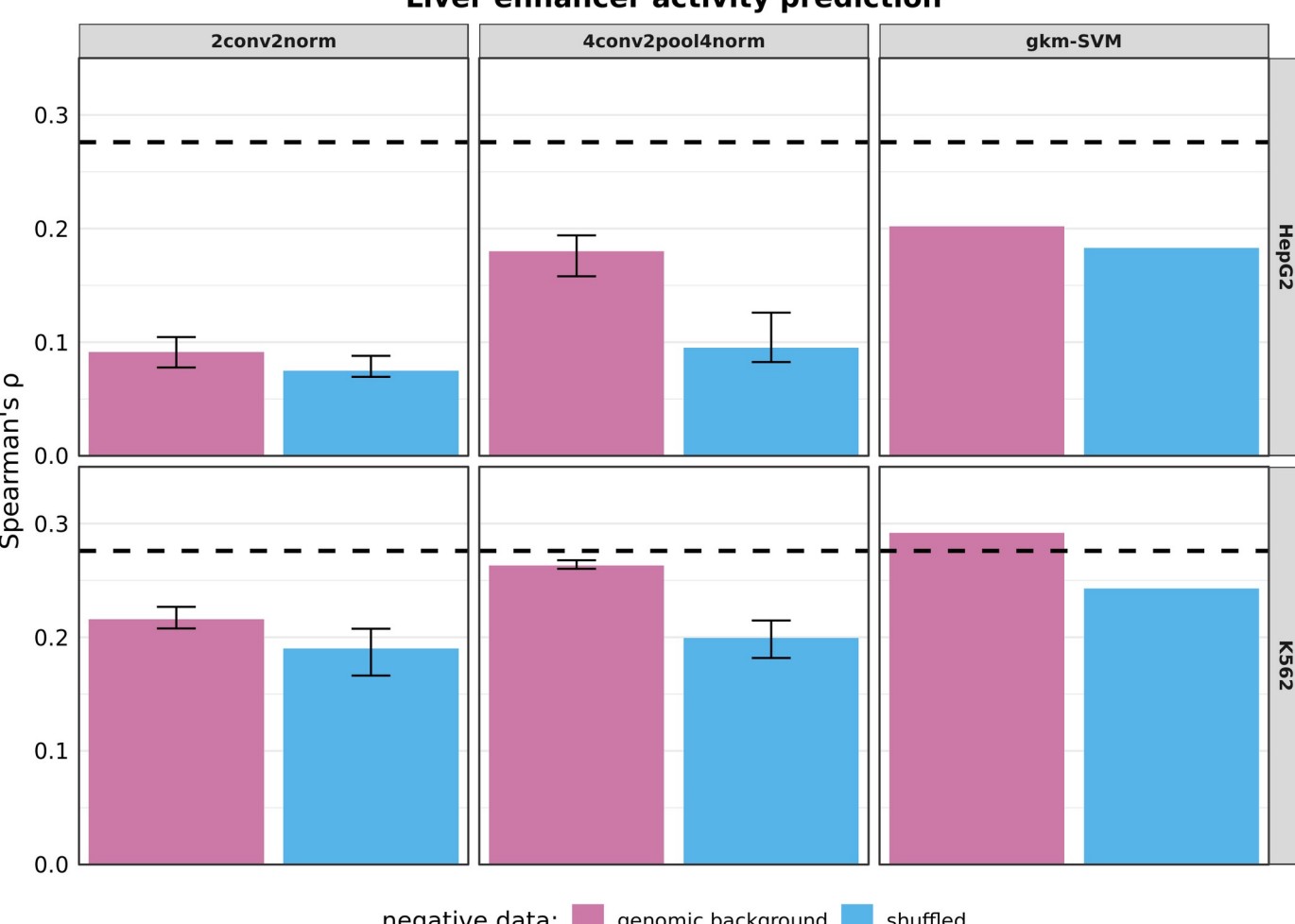

**Fig 3. HepG2 and K562 model performance for enhancer activity prediction.** Models were trained either on DHS sequences active in HepG2 or K562 cells (positive) and negative sequences, where sets are either composed of genomic background ($t_{GC} = 0.1$) or shuffled ($k = 3$) sequences. Models were tested on enhancer sequence activity readouts previously published for HepG2 cells [25]. Spearman rank correlation of predicted scores and $\log_2$ RNA/DNA ratios was used to evaluate model performance. For 2conv2norm and 4conv2pool4norm bars represent the median of multiple model training runs (n = 10) while error bars represent 1st and 3rd quartiles. The dashed black line (Spearman's $\rho$ = 0.276) represents a reference value which was previously achieved [25].

and the smaller 2conv2norm network architecture reduced performance on genomic background sets. These and results of k-mer shuffled negative sets suggest that models primarily learn representation differences of short motifs. We note that we selected all shuffles to minimize the 8-mer overlap with the positive sequence template, i.e. sequences that mutate the overall motif positioning. We could also show that k-mer size is correlated to the number of known TFBS motifs found in the negative training sequences and that shuffled sequences have a higher proportion of rare genomic 8-mers than DHS sequences and genomic background sequences. We suggest that learning rare motifs is the reason that model performance for active DHS prediction seems highest when using highly shuffled sequences (k = {1..3}) as negative training data, but drops considerably when applying models to validation sets using genomic background negative sets. Independent of that effect, genomic background sequences also outperformed shuffles for k higher than 4 for active DHS prediction.

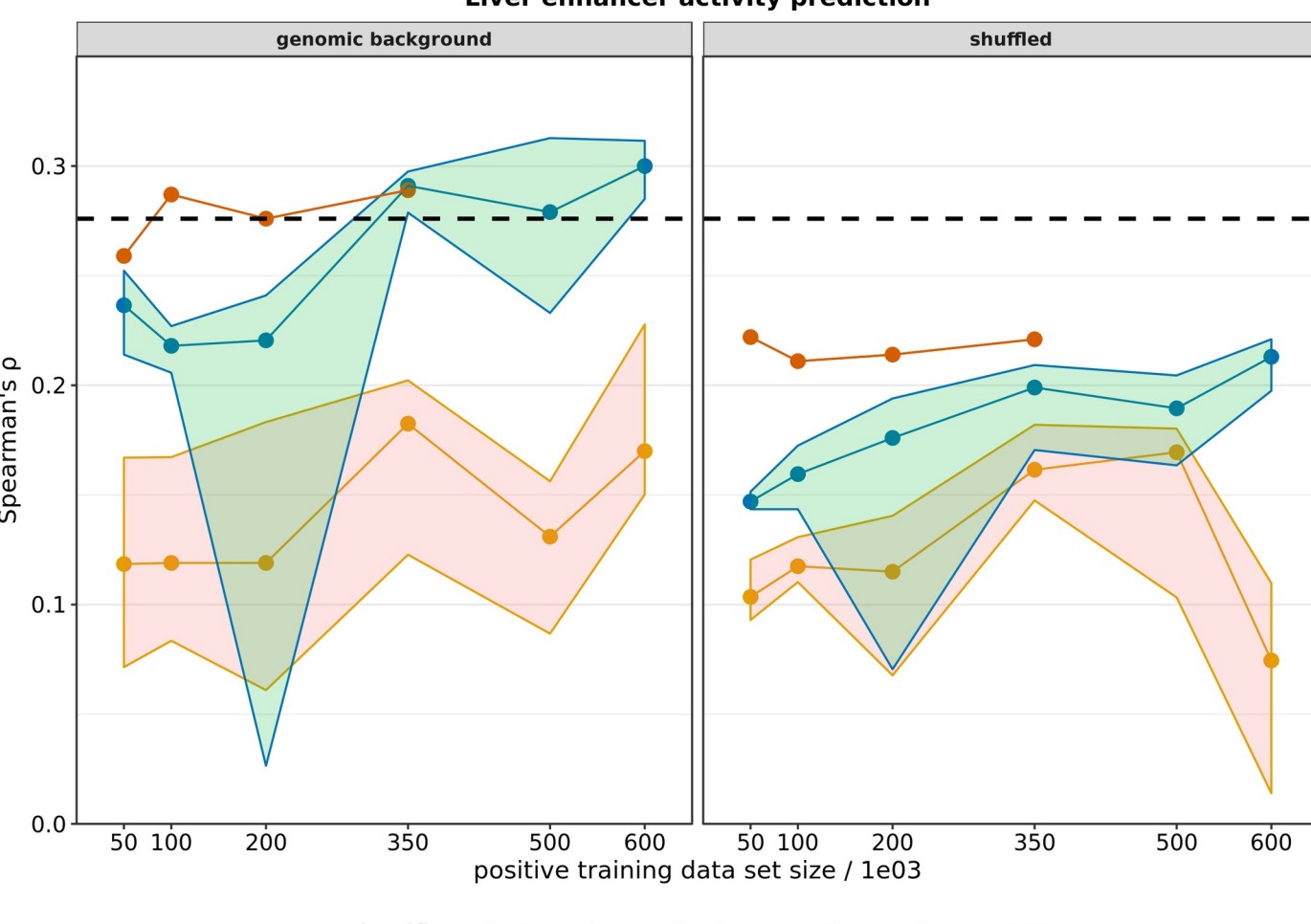

**Fig 4. Model performance in enhancer activity prediction for different training set sizes.** Models were trained on datasets of different sizes composed of DHS sequences (positive) created by sampling of multiple DHS sets of different cell types, and corresponding negative sequence sets, composed of genomic background ($t_{GC}$ = 0.1) (on the left) or shuffled ($k$ = 3) sequences (on the right). Classifiers are represented with different colors. Due to long training durations, gkm-SVM models were trained up to a maximum size of 350k positive samples. Models were tested on enhancer sequences active in HepG2 cells from which activity readouts were previously published [25]. Spearman rank correlation of predicted scores and log2 RNA/DNA ratios was used to evaluate model performance. Dots represent median values of repeated model training (n = 10) while ribbons represent 1st and 3rd quartiles. The dashed black line (Spearman's ρ = 0.276) represents a reference value achieved previously by a gkm-SVM using ENCODE ChIP-seq peaks versus matched control sequences [25].

Since shuffled sequences are artificial and lack biological constraints, models based on this kind of negative set may learn differential sequence motif representations that correspond to genuine TFBS motifs (both active or inactive in the specific cell-type) and differential motif representation due to other biological constraints (e.g. underrepresentation of CpG dinucleotides). While density of binding sites was previously shown to be predictive of regulatory activity [25,46], quantitative and tissue-specific predictions require the models to learn motifs directly related to sequence activity (e.g. active TFBS in a certain cell-type). Consequently, for the two tasks of tissue-specific activity and quantitative activity prediction, genomic background sequences perform always better than sequence shuffles. In line with these observations, models trained on longer preserved k-mers perform better for these tasks, while still falling behind models using the genomic background. We conclude that with background genomic sequences as negative training data, model training tends to ignore patterns present

in natural DNA sequences and is able to focus on more subtle differences in binding site representation.

These patterns are consistent across gkm-SVM and CNN models. On the "complex" tasks, gkm-SVM models outperformed the CNN models in our setup. While we do not see a clear difference between CNN architectures for tissue-specific DHS regions, in the quantitative enhancer activity predictions, the more complex 4conv2pool4norm architecture performs considerably better. For biologically meaningful results, appropriate training datasets are always required and we showed on this last task that training set sizes for CNNs need to be much larger to reach gkm-SVM model performance. The amount of training data is also just one parameter that influences CNN model performance and there are many other network and training hyperparameters that can be tuned.

The quantitative predictions also revealed an issue with the commonly used software package for drawing background sequences from the genome. While in the first two tests, the GC matching parameter did not seem to make a difference, a larger deviation in GC matching provided a performance increase in quantitative enhancer activity prediction. Concurrently, the HepG2 enhancer activity readouts show a positive correlation of GC content with enhancer activity (Spearman $\rho$ of 0.24 with MaxGC feature in the previous publication, [25]). We therefore looked more rigorously at the GC matching and noticed that even for the most stringent setting, high GC-content DHS regions are not sufficiently matched with genomic background sequences (S17 Fig). This causes the models to learn sequence GC content as predictive of regulatory activity rather than specific sequence patterns. We need to highlight a necessary balance in sequence matching attempts though. While trying to compensate for experimental biases in open chromatin data, we might need to acknowledge a real GC signal due to an enrichment of open chromatin in GC-rich active open chromatin regions, like CpG island promoters [47].

We note the GC content might only be one aspect where DHS sites marry multiple functional units, with potentially different characteristics. The bimodality apparent in the GC content of the DHS sites used (S17 Fig) is likely due to CpG island/GC-rich promoters, as others have previously described unimodal distributions for enhancers [48]. This not only argues for good empirical matching of GC characteristics (incl. a long tailed GC distribution of enhancers [48]), but also prompts an argument that DHS datasets could potentially be better modeled if split-up. It has long been described that enhancers and promoters differ in their histone code [49] and that they at least partially interact with different DNA binding proteins [50,51]. However, the lines are blurred and they are overall similar in DNA sequence, chromatin and TF architecture [52,53]. Further, when starting to define classes, subclasses within promoters immediately come into focus. For example, CpG-rich promoters are associated with ubiquitously expressed genes or complex expression patterns (e.g. during embryonic development) [54,55], while CpG-poor promoters are associated with tissue-specific expression [54] and similar to enhancers in terms of recruited TFs [56]. Others have described intragenic enhancers acting as alternative promoters [57] and promoters acting as enhancers of other promoters [58]. This makes a distinction context-dependent [52], something that is not modeled in the CNNs we explored. Even though not explicitly tested in our study, we do not expect genomic background sequences to work better for one type of regulatory sequence but not the other. We identified learners including rare motifs from artificial sequence sets as the main disadvantage of shuffled sequences. As this is linked to the rareness of motifs in the background, this observation is expected to hold for subsets of open chromatin regions. Further, across datasets used here, promoters contribute on average 12% of DHS sites, with GC-rich promoters contributing an even smaller fraction of that. Splitting up DHS peaks into different types of regulatory elements will considerably reduce training data set size, which we identified as a major

bottleneck in training (Fig 4). Together with the biochemical and functional overlaps between these classes, rather than splitting the data, combined Deep Learning models with shared as well as separate layers might be worthwhile to explore in future studies.

## Conclusions

Regulatory sequences are essential for all cellular processes as well as cell-type specific expression in multicellular organisms. A better understanding of the encoding of regulatory activity in DNA sequences is critical and will help to decipher the complex mechanisms of gene expression. Supervised machine learning methods like gkm-SVMs and CNNs can identify associated patterns in DNA sequences [5], however to build the respective models, positive sets of active regulatory sequences and negative sets of inactive sequences are required. We use open chromatin regions as a general proxy for regulatory sequences and do not differentiate between promoters or enhancers/silencers. While proxies for active regions (e.g. DHS open chromatin sites) are widely available for many cell-types and organisms, negative sets are typically computationally derived from genomic background sequences or shuffles of the positive sequences.

To assess whether one approach is preferable over the other, we contrasted both in several experiments. Our results indicate an important influence of negative training data on model performance. Multiple results show that genomic sequences are the better choice for more biologically meaningful results and, when using shuffled sequences, the model performance highly depends on the size of the preserved k-mers.

While k-mer shuffling is computationally efficient and generates synthetic DNA sequences, selection of genomic background sequences involves matching of certain properties of the positive training set (e.g. length, GC content, repeat fraction) which makes it computationally more expensive. With the genomic background method applied here [17], we notice that GC matching should be improved to closely reproduce the continuous GC density distribution of the positive set rather than a mean and standard deviation. Further, for both types of negative sets, it is only assumed that sequences are regulatory inactive. For the shuffles this assumption is based on the artificial nature of sequences, for the background it is based on the excluded overlap with active sequences. While this might generally argue for semi-supervised learning approaches, comprehensive positive sets may somewhat alleviate the issue for genomic background sets.

Comparing two different machine learning approaches, we show that gkm-SVMs give very robust and good results, while CNNs performance could be improved by larger training datasets. This is inline with gkm-SVMs being the simpler machine learning approach (despite being slower in their current implementation [33,59]) and we see this as a cautionary reminder to keep models simple, especially if training data is limited. Apart from the negative training data analyzed here, network architecture and training parameters of CNNs should be explored and optimized in future work. The parameter space of CNNs is immense and remains largely underexplored. Further, multi-task CNN implementations show improved performance [21,60], potentially also due to the effective increase in training data. However, to focus our analysis on the effects of the negative set and to keep comparisons to gkm-SVMs possible, we did not include these here.

To conclude, this study provided relevant insights about how regulatory activity is encoded in DNA sequence, like highlighting the importance of short sequence motifs, and yielded important insights for training machine learning models. We show that negative training data is of high importance for model performance and that the best results are obtained when using sufficiently large and well-matched genomic background datasets. Comparing different

learners, we see that gkm-SVMs are very robust and provide good overall performance. While CNNs have the potential to outperform these simpler models, they require careful attention to the selection of adequate architectures and hyperparameter optimization. While not a focus of this work, models may be further interpreted with respect to their sequence features learned [61,62], in order to shed more light upon the sequence encoding of gene regulation.

## Supporting information

**S1 Fig. Estimated loss on the training and validation sets over training epochs for 2conv2norm models.** Each model was trained on a HeLa-S3 DHS (positive) training dataset and a 2-mer shuffled (negative) training dataset using the 2conv2norm classifier. Training was repeated 10 times and results are represented in different shades of blue while the mean values are represented in orange. Estimated loss in the training set and the validation set are displayed on the left and right, respectively.
(PDF)

**S2 Fig. Estimated loss on the training and validation sets over training epochs for 4conv2pool4norm models.** Each model was trained on a HeLa-S3 DHS (positive) training dataset and a 2-mer shuffled (negative) training dataset using the 4conv2pool4norm classifier. Training was repeated 10 times and results are represented in different shades of blue while the mean values are represented in orange. Estimated loss in the training set and the validation set are displayed on the left and right, respectively.
(PDF)

**S3 Fig. Recall values for regulatory sequence prediction on validation sets of models trained on genomic background sequences.** Each model was trained on a DHS (positive) training dataset and a genomic background (negative) training dataset and tested on a chromosome 21 hold-out validation set. Recall was calculated as a measure of model performance. For each classifier three different negative training sets are compared where the tolerances of differences in GC content composition ($t_{GC}$) is varied. Each model was trained on data derived from one cell line. Bars represent the mean of multiple cell lines and technical replicates (n = 7 for gkm-SVM, n = 70 for CNNs: 10 replicates per cell line) while error bars represent the standard deviation.
(PDF)

**S4 Fig. AUPRC values or regulatory sequence prediction on validation sets of models trained on genomic background sequences.** Each model was trained on a DHS (positive) training dataset and a genomic background (negative) training dataset and tested on a chromosome 21 hold-out validation set. Area under precision recall curve (AUPRC) was calculated as a measure of model performance. For each classifier three different negative training sets are compared where the tolerances of differences in GC content composition ($t_{GC}$) is varied. Each model was trained on data derived from one cell line. Bars represent the mean of multiple cell lines and technical replicates (n = 7 for gkm-SVM, n = 70 for CNNs: 10 replicates per cell line) while error bars represent the standard deviation.
(PDF)

**S5 Fig. Recall values for regulatory sequence prediction on validation sets of models trained on shuffled sequences.** Each model was trained on a DHS (positive) training dataset and a k-mer shuffled (negative) training dataset and tested on a chromosome 21 hold-out validation set. Recall was calculated as a measure of model performance. For each classifier seven different negative training sets are compared where the size of preserved k-mers during

shuffling is varied. Each model was trained on data derived from one cell line. Bars represent the mean of multiple cell lines and technical replicates (n = 7 for gkm-SVM, n = 70 for CNNs: 10 replicates per cell line) while error bars represent the standard deviation.
(PDF)

**S6 Fig. AUPRC values for regulatory sequence prediction on validation sets of models trained on shuffled sequences.** Each model was trained on a DHS (positive) training dataset and a k-mer shuffled (negative) training dataset and tested on a chromosome 21 hold-out validation set. Area under precision recall curve (AUPRC) was calculated as a measure of model performance. For each classifier seven different negative training sets are compared where the size of preserved k-mers during shuffling is varied. Each model was trained on data derived from one cell line. Bars represent the mean of multiple cell lines and technical replicates (n = 7 for gkm-SVM, n = 70 for CNNs: 10 replicates per cell line) while error bars represent the standard deviation.
(PDF)

**S7 Fig. Number of transcription factor binding motifs in training sequences.** Known human transcription factor binding site (TFBS) motifs were matched in training sequences of different datasets from different cell lines (n = 7). Bars represent the mean value, error bars the standard deviation.
(PDF)

**S8 Fig. Recall values for regulatory sequence prediction.** Models were trained on sequences of DHS regions (positive) with corresponding sets of negative sequences and tested on a chromosome 8 hold-out test set. For each classifier two different negative training sets are compared; sequences were either chosen from genomic background ($t_{GC}$ = 0.02) or generated by shuffling positive sequences and preserving k-mer counts (k = 2). Recall was calculated to compare model performance. Seven models were trained on data derived for specific cell lines, bars represent the mean and error bars the standard deviations across models. Pairwise comparisons were performed with Wilcoxon signed-rank tests and asterisks represent significance levels (*p<0.05, **p<0.01, ***p<0.001).
(PDF)

**S9 Fig. AUPRC values for regulatory sequence prediction on test sets.** Each model was trained on a DHS (positive) training dataset and a set of neutral sequences (negative) and tested on a chromosome 8 hold-out test set. Recall was calculated as a measure of model performance. For each classifier two different negative training sets are compared. Sequences were either chosen from genomic background ($t_{GC}$ = 0.02) or generated by shuffling positive sequences and preserving k-mer counts (k = 2). Each model was trained on data derived from one cell line. Bars represent the mean of multiple cell lines (n = 7) while error bars represent standard deviations. Pairwise comparisons were performed with Wilcoxon signed-rank test and asterisks represent significance levels (*p<0.05, **p<0.01, ***p<0.001).
(PDF)

**S10 Fig. Genomic frequency of 8-mers in different classes of the test sets and the first convolutional layer of the CNN models.** Exemplary for all cell-types, the figure shows results for HeLa-S3. Genomic frequency of 8-mers was extracted across all major human chromosomes and Z-Score transformed (i.e. mean-centered and standard deviation normalized to one). Panel (A) shows the genomic frequency of 8-mers in the test sets split out as DHS sites (orange, positive class), negative genomic background sequences (shades of red, from low to high) and different negative k-mer shuffles (shades of blue, from low to high). Smaller k-mer shuffles

contain more rare genomic 8-mers. Panel (B) shows the distribution of the genomic 8-mer frequency for the top 100 sequences for each of 128 kernels in the first convolutional layer for 2conv2norm (left) and 4conv2pool4norm (right) architectures.
(PDF)

**S11 Fig. Pairwise sequence overlap in (a) training, (b) validation and (c) test sets.** We determined the overlap of merged peak sets across experiments in the same cell-type and across cell-types. For peaks to be considered overlapping between datasets, we required a 70% overlap in their coordinate ranges. We calculated pairwise overlap as number of overlapping peaks divided by the number of peaks in the union of both data sets. Datasets are named according to S1 Table.
(PDF)

**S12 Fig. HeLa-S3 model performance for tissue-specific regulatory sequence prediction on validation sets of models trained on genomic background sequences.** Models were trained on DHS sequences (positive) active in HeLa-S3 cells and neutral sequences from genomic background (negative) with varied GC content tolerance ($t_{GC}$). Models were tested on DHS sequences specifically active in HeLa-S3 (positive) and DHS sequences active only in one or multiple other cell lines (A549, HepG2, K562, MCF-7) (negative). (A) and (B) show ROC and PR curves for 2conv2norm models, (C) and (D) show ROC and PR curves for 4conv2pool4-norm models, (E) and (F) show ROC and PR curves for gkm-SVM models. Corresponding AUROC and AUPRC values are included.
(PDF)

**S13 Fig. HeLa-S3 model performance for tissue-specific regulatory sequence prediction on validation sets of models trained on shuffled sequences.** Models were trained on DHS sequences (positive) active in HeLa-S3 cells and neutral sequences from genomic background (negative) with varied size of preserved k-mers. Models were tested on DHS sequences specifically active in HeLa-S3 (positive) and DHS sequences active only in one or multiple other cell lines (A549, HepG2, K562, MCF-7) (negative). (A) and (B) show ROC and PR curves for 2conv2norm models, (C) and (D) show ROC and PR curves for 4conv2pool4norm models, (E) and (F) show ROC and PR curves for gkm-SVM models. Corresponding AUROC and AUPRC values are included.
(PDF)

**S14 Fig. HepG2 model performance for enhancer activity prediction of models trained on genomic background sequences.** Models were trained on HepG2 DHS sequences (positive) and genomic background sequences (negative), where different genomic background sets result from a variation of the GC content tolerance ($t_{GC}$). Models were tested on enhancer activity readouts in HepG2 cells [25]. Spearman rank correlation of predicted scores and log2 RNA/DNA ratios was used to evaluate model performance. Error bars represent 95% confidence intervals.
(PDF)

**S15 Fig. HepG2 model performance for enhancer activity prediction of models trained on shuffled sequences.** Models were trained on HepG2 DHS sequences (positive) and genomic background sequences (negative), where different genomic background sets result from a variation of the size of preserved k-mers. Models were tested on enhancer activity readouts in HepG2 cells [25]. Spearman rank correlation of predicted scores and log2 RNA/DNA ratios was used to evaluate model performance. Error bars represent 95% confidence intervals.
(PDF)

**S16 Fig. Model performance for enhancer activity prediction of A549, HeLa-S3 and MCF-7 models.** Models were trained either on DHS sequences active in A549, HeLa-S3 or MCF-7 cells (positive) and neutral sequences (negative), where different negative sets are composed of genomic background ($t_{GC}$ = 0.1) or shuffled ($k$ = 3) sequences. Models were tested on activity readouts of enhancer sequences in HepG2 cells [25]. Spearman rank correlation of predicted scores and log2 RNA/DNA ratios was used to evaluate model performance. For 2conv2norm and 4conv2pool4norm bars represent the median of multiple replicates (n = 10) while error bars represent 1st and 3rd quartiles. The dashed black line represents a reference value (Spearman's ρ = 0.276) achieved previously [25].
(PDF)

**S17 Fig. Distribution of GC content in sequences of HepG2 training datasets.** The distribution of the sequences' GC contents in a dataset of active DHS regions in HepG2, three corresponding genomic background datasets with varied GC content tolerance ($t_{GC}$) and a set of random 300 bp sequences from the genome is shown.
(PDF)

**S1 Table. Overview of DNase-seq datasets.** The number of DHS sequences is given after merging replicates and exclusion of alternative haplotypes, unlocalized genomic contigs and sequences containing non-ATCG bases. The datasets were split up into training, validation (chromosome 21) and test (chromosome 8) sets. The number of samples in these sets are given in the respective columns. Experiment and Replicate IDs are referring to ENCODE accessions.
(PDF)

**S2 Table. Overview of tissue-specific validation and test sets.** Tissue-specific positive samples are DHS sequences of one cell line not overlapping with DHS sequences of the other cell lines. In contrast, negative samples are DHS sequences of other cell lines not overlapping with the first cell line. For A549, one dataset was chosen (B, named according to S1 Table). For MCF-7 one dataset was chosen (B, named according to S1 Table). The number of DHS sequences is given after exclusion of alternative haplotypes, unlocalized genomic contigs and sequences containing non-ATCG bases. The validation and test sets contain sequences located on chromosome 21 and 8, respectively.
(PDF)

**S3 Table. Layer properties of 4conv2pool4norm network.** The column named 'Size' provides the convolutional kernel size, the max-pooling window size, the relative dropout size and the dense layer size depending on information given in column 'Layer type'.
(PDF)

**S4 Table. Layer properties of 2conv2norm network.** The column named 'Size' provides the convolutional kernel size, the max-pooling window size, the relative dropout size and the dense layer size depending on information given in column 'Layer type'.
(PDF)

**S5 Table. 2conv2norm recall for regulatory sequence prediction for different cell lines.** Ten CNN models of the 2conv2norm architecture were trained each on DHS datasets (positive) and corresponding negative sets of k-mer shuffled sequences ($k$ = 2, $k$ = 7) or genomic background sequences ($t_{GC}$ = 0.02) for A549 or MCF-7 cells. A549 and MCF-7 cell lines are represented in our data with two training datasets each, which are labeled as A and B, respectively. Model performance was evaluated based on recall for hold-out sets (chromosome 8). The table summarizes mean and standard deviation across ten trained models. There are seven different hold-out sets derived from different cell lines and we assess model generalization across cell-

types. Datasets are named according to S1 Table. Respective results for the gkm-SVM models are available Table 1, results for CNN models of 4conv2pool4norm architecture are available in S6 Table.

(PDF)

**S6 Table. 4conv2pool4norm recall for regulatory sequence prediction for different cell lines.** Ten CNN models of the 4conv2pool4norm architecture were trained each on DHS datasets (positive) and corresponding negative sets of k-mer shuffled sequences (k = 2, k = 7) or genomic background sequences ($t_{GC}$ = 0.02) for A549 or MCF-7 cells. A549 and MCF-7 cell lines are represented in our data with two training datasets each, which are labeled as A and B, respectively. Model performance was evaluated based on recall for hold-out sets (chromosome 8). The table summarizes mean and standard deviation across ten trained models. There are seven different hold-out sets derived from different cell lines and we assess model generalization across cell-types. Datasets are named according to S1 Table. Respective results for the gkm-SVM models are available Table 1, results for CNN models of 2conv2norm architecture are available in S5 Table.

(PDF)

**S7 Table. AUROC values for tissue-specific regulatory sequence prediction on validation sets.** Models were trained on DHS sequences (positive) with corresponding sets of negative sequences and tested on a set of tissue-specific chromosome 21 test set. For each classifier two different negative training sets are compared; sequences were either chosen from genomic background ($t_{GC}$ = 0.1) or generated by shuffling positive sequences and preserving k-mer counts ($k$ = 7). AUROC value was calculated to compare model performance.

(PDF)

**S8 Table. AUPRC values for tissue-specific regulatory sequence prediction on validation sets.** Models were trained on DHS sequences (positive) with corresponding sets of negative sequences and tested on a set of tissue-specific chromosome 21 test set. For each classifier two different negative training sets are compared; sequences were either chosen from genomic background ($t_{GC}$ = 0.1) or generated by shuffling positive sequences and preserving k-mer counts ($k$ = 7). AUPRC value was calculated to compare model performance.

(PDF)

## Acknowledgments

We thank current and previous members of the Kircher group for helpful discussions and suggestions. Specifically, we would also like to acknowledge input from Giorgio Valentini and his lab at Università degli Studi di Milano, as well as Dirk Walther at the University of Potsdam. Computation has been performed on the HPC for Research cluster of the Berlin Institute of Health.

## Author Contributions

**Conceptualization:** Max Schubach, Martin Kircher.

**Data curation:** Louisa-Marie Krützfeldt.

**Formal analysis:** Louisa-Marie Krützfeldt, Max Schubach, Martin Kircher.

**Funding acquisition:** Martin Kircher.

**Investigation:** Louisa-Marie Krützfeldt, Max Schubach, Martin Kircher.

**Methodology:** Louisa-Marie Krützfeldt.

**Project administration:** Max Schubach, Martin Kircher.

**Supervision:** Max Schubach, Martin Kircher.

**Visualization:** Louisa-Marie Krützfeldt.

**Writing – original draft:** Louisa-Marie Krützfeldt.

**Writing – review & editing:** Max Schubach, Martin Kircher.

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
