## [Decision Letter · Decision Letter 0]

18 Aug 2020

PONE-D-20-22794

The impact of different negative training data on regulatory sequence predictions

PLOS ONE

Dear Dr. Kircher,

Thank you for submitting your manuscript to PLOS ONE. After careful consideration, we feel that it has merit but does not fully meet PLOS ONE’s publication criteria as it currently stands. Therefore, we invite you to submit a revised version of the manuscript that addresses the points raised during the review process.

As you can appreciate from the attached reports, both reviewers agreed that this is an important study, and that is was generally well conducted, but they have made suggestions for improving certain aspects. In particular, we feel that it would be important to address the major points raised by reviewer #2 regarding the mixing of promoters and enhancers in the analyses, and performing cross-cell line tests using a genomic background negative set.

We look forward to receiving your revised manuscript.

Kind regards,

Miguel Branco

Academic Editor

PLOS ONE

Journal Requirements:

2.Thank you for stating the following in the Acknowledgments Section of your manuscript:

[This work was supported by the Berlin Institute of Health and Charité – Universitätsmedizin

Berlin. The funder had no involvement in study design; in the collection, analysis and

interpretation of data; in the writing of the report; and in the decision to submit the article for

publication.]

 [The author(s) received no specific funding for this work.]

Reviewers' comments:

Reviewer's Responses to Questions

**Comments to the Author**

1. Is the manuscript technically sound, and do the data support the conclusions?

Reviewer #1: Yes

Reviewer #2: Yes

2. Has the statistical analysis been performed appropriately and rigorously? 

Reviewer #1: Yes

Reviewer #2: Yes

3. Have the authors made all data underlying the findings in their manuscript fully available?

Reviewer #1: Yes

Reviewer #2: Yes

4. Is the manuscript presented in an intelligible fashion and written in standard English?

Reviewer #1: Yes

Reviewer #2: Yes

5. Review Comments to the Author

Reviewer #1: Krutzfeldt et al. investigate 3 machine learning models (2 CNN and one gkm-SVM) on predicting accessible regions and enhancers in human cell lines from DNA sequence alone. In particular, the authors focused on selection of different negative sequences and how that affects the model performs. The authors are able to identify with high accuracy accessible regions of the genome. Nevertheless, their results are relatively poor for cell specific accessible regions and enhancers, which is expected for DNA sequence only models. Furthermore, the authors show that using one model to select the negative set (genomic background) provides better and robust results in most test scenarios. The paper is well written and presents important results. There are some points that authors would need to address before I could recommend this paper for publication:

1. An explanation of CNNs and gkm-SVMs could help readers that are unfamiliar with these machine learning algorithms.

2. Line 66-74, the authors mention that there is a strong link between TF binding and DNA accessibility. While this would be the case for many TFs, it is not generally true (see https://www.sciencedirect.com/science/article/abs/pii/S1097276520303579?via%3Dihub, https://science.sciencemag.org/content/368/6498/1460 or https://www.biorxiv.org/content/10.1101/666446v1.abstract). I think the authors should discuss about these different types of mechanisms.

3. Line 75-85, the DNA sequence can predict TF binding sites, but many TFs bind in a concentration dependent manner (e.g. https://journals.plos.org/plosgenetics/article?id=10.1371/journal.pgen.1001290, https://journals.plos.org/plosgenetics/article?id=10.1371/journal.pgen.1003571

or https://academic.oup.com/nar/article/43/1/84/2903035).

4. “The tolerance for differences in repeat ratio and relative sequence length were set to 0, but the tolerance for differences in GC content was varied for different training datasets (tGC={0.02, 0.05, 0.1}).” – The authors need to explain why this was varied.

5. “The parameter k which indicates the size of the preserved k-mers was varied for different datasets (k=[1,7])” – Again, it the authors need to explain why this was varied.

6. For liver enhancer activity data, have the enhancers from Inoue et al. (2017) been cross validated by another method? For example with STARR-seq from https://genomebiology.biomedcentral.com/articles/10.1186/s13059-018-1473-6 .

7. “We evaluated models on their respective hold-out and additionally the models trained on shuffled data on hold-out using genomic background sequences as negative sets” – This makes sense after reading and re-reading it a few times but it isn’t immediately clear. I’d recommend finding a way to re-word it.

8. It wasn’t particularly clear exactly what information the authors use as an input. In Min et al. (2016) they use a 300bp sequence as an input. As they are using DeepEnhancer and a smaller network based on the DeepEnhancer architecture I assumed this was the same, but then I wondered how you dealt with merging of positive bins. In the materials and methods section you state that positive DNA-seq datasets were created where: “Multiple technical replicates were merged into one file per experiment, combining overlapping (minimum of 1 bp) or adjacent sequences into a single spanning sequence”. When two regions are merged, is a new centre is calculated between the two, then the input region is classed as 300bp around the new centre?

9. Do you treat a peak within 150bp of the centre of another peak as a separate instance? This peak is not overlapping and is not adjacent, but it would contain some of the same sequence.

10. Citations for some Biocoductor packages are missing. Cite the recommended papers.

11. Line 233: “First models were tested on validation sets to identify best parameters for generating the negative training set based only on recall measures.”. Maybe I misunderstood something, but shouldn’t the training be done on the training set not the validation set?

12. “However, models trained on highly shuffled data perform significantly better than models trained on genomic background data; potentially the result of an improper evaluation on varying compositions of the validation sets using different negative data” – Have the authors compared the compositions of the validation sets to test this?

13. When mentioning Figure S10, A and B are mentioned in the text but not C. C is the Z-score of the larger CNN (which looks much the same as B) but it’s just worth pointing out.

14. Is there generalisation data for the CNNs as well as the Gkm-SVM models? (as in Table 1)

15. For S5 and S6 probably shouldn’t be mentioned until after it is explained why k=7 is used. After the previous section explaining that k=2 was close to optimal it seems strange to jump to the new figure without the explanation that comes further down the paragraph.

16. It isn’t particularly clear how models were trained on enhancers. Are there annotated datasets for those cell lines?

17. As a general comment, it is easier to read Figure S1 rather than S1 Figure.

18. Line 305: I would reference https://doi.org/10.1016/j.tig.2009.08.003 about TF motif size in eukaryotic genomes, which could explain why 7-Kmer would overlap with many TF motifs.

19. K-mer 1 is the best, but authors use K-mer 2. I think the authors should include K-mer 1 in their manuscript in order to allow the reader to judge the performance of the model.

20. Figure 1 needs better explanation. It took me a while to understand the difference between top and bottom panels and red/blue colours.

21. Lines 338-348: needs to be explained better. It was difficult to follow.

22. In Figure S10A, k-mer shuffle plots several blue lines; they look similar with DHS and genomic background except for one outlier. Can authors clarify this?

23. Table 1: The authors need to show a Venn diagram with the overlap of DHS between the different cell lines, so the reader can judge these results. The reason the model generalises so well might be because the DHS in the different cells overlap significantly.

24. Lines 397-404: The authors need to discuss that the reason the model doesn’t perform so well on cell specific data is because while DNA sequence is the same, there something else that controls tissue specific regions. Their method is better at predicting constitutive DHS and enhancers and will suffer when predicting tissue specific enhancer.

25. Authors should include a graph with the run time of the different models to allow quantitative assessment of the model.

26. It is not clear why for different cell types different optimal K-mers are used. Can the authors explain this?

27. Lines 431-437: I assumed that the data is available only for HepG2, but what data is used for K562, A549 and HeLa?

28. The performance to predict enhancers is relatively poor, but within the results of other papers. The authors should make this clear.

Reviewer #2: In the manuscript by Krutzfeldt et al., the authors assess the impact of different negative training sets on regulatory sequence prediction. Three models (one gkm-SVM and two CNN) are trained using DNase hypersensitive sites as positive set and either genomic background or shuffled sequences as negative set. Effects of the choice of background are compared across three tasks: hold-out prediction regulatory sequences, cell type specific hold-out prediction and prediction of enhancer activity (correlation with experimental data). The authors find that shuffled sequences contain rare/artificial 8-mers/motifs which are used by the learners which makes models perform worse on biologically more relevant tasks.

This is a wonderful example of how the definition of ‘background’ (i. e. the baseline you compare whatever you are interested in against) has a large influence on the results – which is a common theme in bioinformatics. Unfortunately, the question which background to choose is often not given enough thought. Therefore, studies like this are very relevant and can be eye opening. With machine learning methods becoming more accessible and more widely used information on what to watch out for is pertinent. The manuscript contains relevant data, the rationale is well explained and methods are sufficiently described. It is written in an easy to follow style.

Major points for consideration

It seems likely that looking at all DHS sites together results in a mix of functional biological elements, presumably dominated by enhancers and promoters. These are likely defined by different sequence features and we could envision a scenario in which promoter activity might be strongly influenced by presence/absence of a TFBS (strong sequence feature), while enhancer activity might be more dependent on histone modifications (weak sequence feature). Thus, it seems possible that the method of negative region definition (shuffling or genomic background) may impact differently on different functional elements. If this were the case, it would affect interpretation of the results at several points throughout the manuscript. The authors mention one example in the context of GC matching (p18, lines 503-516, Figure S16): The distribution of GC content of the DHS sites is clearly bimodal. It seems likely that the less CG rich peak mainly contains enhancers and non-CGI promoters while the GC rich peak is dominated by CGI promoters. Sequence features learned by a model would probably be very different if learned on those three subgroups and could be affected quite differently by choice of negative set. I am wondering if a very crude distinction of DHS sites into promoter distal (putative enhancers) and CGI and non-CGI promoter proximal would result in substantially improved models and how these would be affected by different negative data.

P 15 line 358, Table 1: In previous paragraphs, the authors have convincingly shown that shuffled 2-mers contain rare/artificial motifs and that the learners are affected by these. Here, these models based on shuffled regions are used to compare performance across cell types. However, since part of the learning is based on the distinction between biological and artificial, performance of a model trained on one cell type is expected to be high in another (we assume that sequence composition doesn’t change between cell lines). How does model performance generalise across cell lines when genomic background is used as negative sets? Does this still support the idea that organismal rather than tissue specific regulatory features are predicted?

The point that models are strongly influenced by differences between biological and artificial sequences is really driven home by the data for tissue-specific regulatory region prediction: models trained on highly shuffled sequences are only marginally better than random (or not at all if k=1!). The comparison above therefore also needs to include data for higher k-mers (7-mers to be able to compare with the figures on tissue specific region prediction). I consider the comparison across cell-types for models trained on genomic background and 7-mer shuffles essential.

P16 line 306: Quantitative enhancer activity prediction: As mentioned above the DHS sites are a mix of functional elements, presumably mainly enhancer and promoter sequences. Therefore, the model will learn both enhancer and promoter sequence features, however, the experimental readout is only enhancer activity. This is bound to impact negatively on correlation, so maybe it’s not so surprising that the correlation is so low. The differences in overall correlation when using different parameters (flexibility in GC content and k-mers) are very small and often the Spearman’s rank correlation hovers around 0.1. This is such low correlation and such small changes, that I would be very reluctant to assign meaning to parameters different to before performing better in this task. This should be made clearer in the text. Moreover, it would be much more informative to plot the actual value pairs rather than just the value for the overall correlation. It might even reveal subgroups of value pairs, some with better correlation than others. An idea might be to colour promoter proximal/distal or CGI/non-CGI data points differently to see if patterns emerge.

Why were chromosome 21 and 8 chosen as validation and test sets, and why was this preferred over randomly selected positive and negative regions? Genomes are known to show chromosomal bias for genomic features and potential differences between training and test/validation chromosomes should be assessed, e. g. sequence composition, density of positive and negative regions, GC content, density of repetitive elements etc.

Minor points

P 13, line 257, Frequency distribution of 8-mers. While all other genomic sequences are based on GRCh38, this frequency analysis uses GRCh37 – presumably because it was reused from a previous study. Since these are both mature genome builds, it is probably fair to assume that the distribution doesn’t change significantly, but do the authors have any information on this? Also, the models include the sex chromosomes, while the frequency analysis doesn’t. Does this matter?

P 14 line 314 “potentially the result of an improper evaluation on varying compositions of the validation sets using different negative data.” I don’t understand the meaning of this sentence.

While explanations are in the text, I found Figure 1 quite confusing and clearer labels would help. The legend title should be something like “model trained on” as pink indicates that the model was trained on genomic background and blue indicates that the model was trained on shuffled sequences. Likewise, the labels for top and bottom panels should include more information, e. g. “negative regions in the test set: genomic background”.

P 14 lines 338 to 348 and Figure S10A: In my mind, this figure contains a key message and one could think about moving it into the main figures. It should be indicated which blue curve belongs to which k-mer and a number/percentage of how many 8-mers were excluded (because they weren’t present in the genome) should be given.

P16 line 431: While surely obvious to the authors, I initially found it confusing that the model from the previous publication performed so much better in the experimental activity prediction. It would probably help the reader to very briefly recap that the model from publication 23 was trained on a corresponding set of ChIPped regions rather than DHS sites.

6. PLOS authors have the option to publish the peer review history of their article (what does this mean?). If published, this will include your full peer review and any attached files.

Reviewer #1: **Yes: **Dr Radu Zabet

Reviewer #2: **Yes: **Christel Krueger

---

## [Author Response · Author response to Decision Letter 0]

5 Oct 2020

We provide our responses as a separate PDF of the submission.

---

## [Decision Letter · Decision Letter 1]

27 Oct 2020

PONE-D-20-22794R1

The impact of different negative training data on regulatory sequence predictions

PLOS ONE

Dear Dr. Kircher,

Thank you for submitting your manuscript to PLOS ONE. Some minor points were raised during the review process that we feel would be important to address. Therefore, we invite you to submit a revised version of the manuscript.

Specifically, given the amount of discussion triggered by the reviewers' comments on the possible impact of not distinguishing between promoters and enhancers, we feel that it would be important for the authors to more extensively present their rationale and views on this matter (during results interpretation and/or in the discussion section). You may also choose to address additional comments from the reviewers.

We look forward to receiving your revised manuscript.

Kind regards,

Miguel Branco

Academic Editor

PLOS ONE

Reviewers' comments:

Reviewer's Responses to Questions

**Comments to the Author**

1. If the authors have adequately addressed your comments raised in a previous round of review and you feel that this manuscript is now acceptable for publication, you may indicate that here to bypass the “Comments to the Author” section, enter your conflict of interest statement in the “Confidential to Editor” section, and submit your "Accept" recommendation.

Reviewer #1: All comments have been addressed

Reviewer #2: (No Response)

2. Is the manuscript technically sound, and do the data support the conclusions?

Reviewer #1: Yes

Reviewer #2: Yes

3. Has the statistical analysis been performed appropriately and rigorously? 

Reviewer #1: Yes

Reviewer #2: Yes

4. Have the authors made all data underlying the findings in their manuscript fully available?

Reviewer #1: (No Response)

Reviewer #2: Yes

5. Is the manuscript presented in an intelligible fashion and written in standard English?

Reviewer #1: Yes

Reviewer #2: Yes

6. Review Comments to the Author

Reviewer #1: The authors addressed majority of the points I raised.

1. Point 3 and 4. We are happy with the changes made by the authors. Nevertheless, the suggested papers to be citated were the minimum and we were expecting the authors to cite those papers we suggested and others.

2. Point 11, the authors explained that part, but did not change the text. Readers might have the same question and I think it is important to be clear.

3. Point 18. The authors mentioned that they do not think the suggested reference is appropriate, but did not provide any explanation for that or any alternative paper.

Reviewer #2: Major Point 1

R: It seems likely that looking at all DHS sites together results in a mix of functional biological elements, presumably dominated by enhancers and promoters. These are likely defined by different sequence features and we could envision a scenario in which promoter activity might be strongly influenced by presence/absence of a TFBS (strong sequence feature), while enhancer activity might be more dependent on histone modifications (weak sequence feature).

A: We appreciate the reviewers' comment and totally agree that these differences may or are even likely to exist. We want to highlight though, that our manuscript focuses on a relative comparison and that we are not attempting to create a superior model for the prediction of open chromatin regions, enhancers or promoters. We use the prediction of open chromatin as a mere example of a relevant prediction task. As mentioned also to the other reviewer, we have not performed an extensive hyperparameter search, nor did we explore multi-task or multi-modal models which are expected to show an increased performance for the mentioned prediction tasks.

R: I completely appreciate that the manuscripts main focus is not on creating the best possible model for open chromatin prediction. The fact remains though that it is possible that background and parameter choice has a different influence depending on what it is that one is trying to predict – and open chromatin regions are clearly a mixed bag. While it may be beyond the scope of this study to look at subgroups of open chromatin regions (this could for example have been done on the first task only to shine some light on how much influence this might have), I find it disappointing that the authors chose to not even discuss this question.

Major point 2 has been addressed.

Other points:

R: P16 line 306: Quantitative enhancer activity prediction: As mentioned above the DHS sites are a mix of functional elements, presumably mainly enhancer and promoter sequences. Therefore, the model will learn both enhancer and promoter sequence features, however, the experimental readout is only enhancer activity. This is bound to impact negatively on correlation, so maybe it’s not so surprising that the correlation is so low. The differences in overall correlation when using different parameters (flexibility in GC content and k-mers) are very small and often the Spearman’s rank correlation hovers around 0.1. This is such low correlation and such small changes, that I would be very reluctant to assign meaning to parameters different to before performing better in this task. This should be made clearer in the text. Moreover, it would be much more informative to plot the actual value pairs rather than just the value for the overall correlation. It might even reveal subgroups of value pairs, some with better correlation than others. An idea might be to colour promoter proximal/distal or CGI/non-CGI data points differently to see if patterns emerge.

A: We assume that the reviewer refers to p18, lines 406ff. We agree with the reviewer that there is a general agreement that Pearson correlations below 0.5 (R2 of 0.25) are not considered predictive models. This is based on a "variance explained by a linear model" argument. However, this does not mean that significant (Spearman) correlations below this value can not be interpreted, or that differences in correlation values have no meaning. Due to the simplicity of how correlations are calculated, they tend to be rather stable in these contexts and the relative performance of models can be evaluated using rank correlations (Spearman rather than Pearson). You can see the low variance observed in calculating Spearman correlation between 10 model training instances in Fig 3. We also like to point out that the publication we are comparing to reports the Spearman correlation of 0.276 (R2 of 0.076). As it is possible to calculate confidence intervals for these values, we have added these confidence levels to Figures S14 and S15 (renamed from S13 and S14 due to the insertion of a supplementary figure). You can see that confidence intervals are different from 0 for most of the model parameters tested and that obtained correlation coefficients, despite being small, are significantly different between certain parameter choices.

R: I agree with the authors that a correlation between prediction and experimental readout does not have to be linear to be useful – I have no problem with using rank correlation here. But I do agree with Reviewer 1 that the correlation is poor (significance is easily reached here because of the numbers) and that this is not made sufficiently clear in the text. Instead, as I mentioned in my last comment, a lot of emphasis is placed on the minute effects of different parameters on the (very) weak correlation. I am fully aware that enhancer predictions based on sequence alone are generally not great, and that making the best possible prediction is not the aim of this study. Again, I find it disappointing that the authors missed the opportunity to include more information on the relationship between prediction and experimental readout – for example like the scatterplot of (ranked) prediction vs experimental readout I suggested previously.

R: Something that I noted when re-reading the manuscript is that at two points in the text relating to Figure 4 it is suggested that gkm-SVM models were trained up to a maximum set size of 350 K, however, the last data point is at 200 K. This should be corrected.

7. PLOS authors have the option to publish the peer review history of their article (what does this mean?). If published, this will include your full peer review and any attached files.

Reviewer #1: No

Reviewer #2: **Yes: **Christel Krueger

---

## [Author Response · Author response to Decision Letter 1]

10 Nov 2020

With this revision, we have added a section discussing the mixed nature of DNase hypersensitive (DHS) sites in terms of enhancer and promoter composition. This new section discusses classes and subclasses among regulatory regions and that enhancers and promoters generally share many features. We eventually conclude that rather than splitting up data, Deep Neural Network models with specific architectures (i.e. a combination of shared and separate layers) may offer some advantage when modeling DHS data and should be explored in future work. In addition, we argue why our results (rare motifs being learned from shuffled backgrounds and insufficient GC matching) are universal with respect to promoter and enhancer subsets of DHS sites. In response to a discrepancy noticed by reviewer 2, we added a data point previously missing from Figure 4. We attach our detailed responses as a separate document to this submission.

---

## [Editor Report · Decision Letter 2]

12 Nov 2020

The impact of different negative training data on regulatory sequence predictions

PONE-D-20-22794R2

Dear Dr. Kircher,

We’re pleased to inform you that your manuscript has been judged scientifically suitable for publication and will be formally accepted for publication once it meets all outstanding technical requirements.

Kind regards,

Miguel Branco

Academic Editor

PLOS ONE

---

## [Editor Report · Acceptance letter]

17 Nov 2020

PONE-D-20-22794R2 

The impact of different negative training data on regulatory sequence predictions 

Dear Dr. Kircher:

I'm pleased to inform you that your manuscript has been deemed suitable for publication in PLOS ONE. Congratulations! Your manuscript is now with our production department. 

Kind regards, 

on behalf of

Dr. Miguel Branco 

Academic Editor

PLOS ONE